# Large Scale Mapping of Indoor Magnetic Field by Local and Sparse Gaussian Processes

**Iad ABDUL-RAOUF**[1,2]   **Vincent GAY-BELLILE**[1]   **Cyril JOLY**[2]   **Steve BOURGEOIS**[1]
**Alexis PALJIC**[2]

[1]Université Paris-Saclay, CEA, List, F-91120, Palaiseau, France

[2]Centre de Robotique, MINES ParisTech, Université PSL, 75006 Paris, France

`iad.abdulraouf@cea.fr`

**Abstract:** Magnetometer-based indoor navigation uses variations in the magnetic field to determine the robot's location. For that, a magnetic map of the environment has to be built beforehand from a collection of localized magnetic measurements. Existing solutions built on sparse Gaussian Process (GP) regression do not scale well to large environments, being either slow or resulting in discontinuous prediction. In this paper, we propose to model the magnetic field of large environments based on GP regression. We first modify a deterministic training conditional sparse GP by accounting for magnetic field physics to map small environments efficiently. We then scale the model on larger scenes by introducing a local expert aggregation framework. It splits the scene into subdomains, fits a local expert on each, and then aggregates expert predictions in a differentiable and probabilistic way. We evaluate our model on real and simulated data and show that we can smoothly map a three-story building in a few hundred milliseconds.

**Keywords:** Gaussian process regression, magnetic field maps, indoor localization

## 1   Introduction

Indoor positioning for robotics is an active and challenging field of research [1, 2]. While vision-based approaches [3] are the most common to address this issue, they struggle in environments with low or repetitive textures and when visual cues change over time [4]. Recently, a novel and promising approach has emerged that leverages the spatial anomalies of the indoor ambient magnetic field. [5]. These anomalies are sufficiently distinct to differentiate between locations [6] and stable enough to ensure that magnetic field maps remain valid over the years [4].

Indoor magnetic-field-based localization solutions, such as [4] built upon extended Kalman filtering and [1] built upon particle filtering, require a map of the field values at any position, and uncertainty estimation. Kalman filters additionally assume a differentiable observation model. Classic Gaussian Process (GP) regression [7] from magnetometer measurements can achieve a differentiable and probabilistic representation. However, it has a $\mathcal{O}(N^3)$ computational complexity, with $N$ the number of data points, which becomes intractable for large datasets.

To address this, Vallivaara et al. [8] have downsampled the training data and then learned map values using only training data in a small radius. Downsampling is suboptimal, and this local approach is discontinuous [9]. Instead, Solin et al. [10] and Menzen et al. [11] proposed sparse approximations of the GP. While these approaches scale well with increasing data density, they perform less effectively with increasing area size [12, 13]. Besides, to date, the fastest sparse magnetic map [11] is built upon structured kernel interpolation [14], which may produce discontinuous predictions [9].

This work proposes a new approach to model the magnetic field that scales with increasing area size and data density. It also results in differentiable magnetic predictions that are efficiently computed, thus being compatible with any localization algorithm. We propose splitting the scene into subdomains and learning a Deterministic Training Conditional sparse GP [15] that has been modified to include prior knowledge from Maxwell equations. Then, neighboring experts' predictions

8th Conference on Robot Learning (CoRL 2024), Munich, Germany.

are smoothly and efficiently aggregated into what we call a Local Bayesian Committee Machine, derived from the classic Bayesian Committee Machine [16] in a principled way.

We thoroughly illustrate our method with several toy examples. We also compare it against the state of the art, on real and simulated data. Evaluations are performed regarding mean square error, mean standardized log loss, smoothness, and runtimes. In particular, we can learn a magnetic map in a three-story building from a trajectory of one kilometer in less than 100 milliseconds.

## 2   Background on Gaussian Process Regression

This section introduces the necessary background on Gaussian process regression, sparse approximations, and local expert aggregations. Computational complexities are discussed with respect to data input $x$ density and domain volume instead of the usual dataset size $N$ and number of latent variable $M$. It brings a new light on the scalability of each model.

### 2.1   Full Gaussian Process Regression

Let us consider a vector field $f(x) \in \mathbb{R}^d$ defined for any $x \in \Omega \subset \mathbb{R}^{d'}$. It must be estimated from a training dataset $\mathcal{D} = \{(x_1, y_1), \ldots, (x_N, y_N)\}$, where each observation $y_i$ of $f(x_i)$ (noted $f_i$) is corrupted by a Gaussian additive white noise. A classic approach is to model $f$ as a zero mean Gaussian process indexed by $x$ [7]

$$f \sim \mathcal{GP}(0, \kappa), \quad y = f(x) + \epsilon, \quad \epsilon \sim \mathcal{N}(0, \sigma_{\text{noise}}^2 I_d), \tag{1}$$

where $\kappa$ is the $d \times d$ covariance kernel such that $\kappa(x, x') = \text{cov}(f(x), f(x'))$. The choice of a specific covariance function encodes the a priori knowledge about the underlying process. It is often defined with respect to hyperparameters, which are learned jointly with $\sigma_{\text{noise}}$ from data or manually tuned to sensible values. For any test input $x_*$, the prior and posterior of $f(x_*)$ (noted $f_*$) are Gaussians and admit a closed-form expression

$$p(f_*) = \mathcal{N}(0, K_{f_*,f_*}), \quad p(f_*|\mathcal{D}) = \mathcal{N}(K_{f_*,\mathbf{f}} \Sigma \mathbf{y}, K_{f_*,f_*} - K_{f_*,\mathbf{f}} \Sigma K_{f_*,\mathbf{f}}^\top) \tag{2}$$

where $\Sigma = (K_{\mathbf{f},\mathbf{f}} + \sigma_{\text{noise}}^2 I_{nd})^{-1}$. The vectors $\mathbf{y}$ and $\mathbf{f}$ are the concatenation $(y_1^\top, \ldots, y_n^\top)^\top$ and $(f_1^\top, \ldots, f_n^\top)^\top$ respectively. We also used the standard shorthand notation $K_{a,b} = \text{cov}(a, b)$. The covariances $K_{\mathbf{f},\mathbf{f}}$ and $K_{f_*,\mathbf{f}}$ are matrices of size $Nd \times Nd$, and $d \times Nd$ respectively. They are defined by block via the covariance kernel (e.g. $\text{cov}(f_i, f_j) = \kappa(x_i, x_j)$).

Training the model is the pre-computation of all quantities in (2) that depend on training data only. Training computational complexity is dominated by the matrix inversion in $\mathcal{O}(d^3 N^3)$ operations. Then a prediction requires $\mathcal{O}(d^3 N^2)$ operations for each input $x_*$. We find it helpful to rewrite the complexities as a function of the volume $|\Omega|$ and the number of points per unit volume $\rho_x = N/|\Omega|$. Training and test complexities respectively become $\mathcal{O}(d^3 \rho_x^3 |\Omega|^3)$ and $\mathcal{O}(d^3 \rho_x^2 |\Omega|^2)$. It is intractable for dense data inputs or in a large environment.

### 2.2   DTC Approximation

To tackle the intractability of exact regression, a large portion of the literature describes sparse approximations [9, 17]. We overview the deterministic training conditional (DTC) sparse approximation formalized by [15]. The starting point is a set of *latent variables* $\mathbf{u} = (u_1^\top, \ldots, u_M^\top)^\top$ called *inducing variables* in the literature, where $M \ll N$. They are values of the GP (like any $f_i$ and $f_*$) at the latent inputs $z_1, \ldots, z_M \in \mathbb{R}^{d'}$. The latent variables are involved in two successive approximations of the joint prior.

$$p(f_*, \mathbf{f}, \mathbf{u}) = p(f_*, \mathbf{f}|\mathbf{u}) p(\mathbf{u}) \simeq p(f_*|\mathbf{u}) p(\mathbf{f}|\mathbf{u}) p(\mathbf{u}) \tag{3}$$

is a conditional independence assumption between the test variable $f_*$ and training variables $\mathbf{f}$. The training conditional is further approximated

$$p(\mathbf{f}|\mathbf{u}) = \mathcal{N}(K_{\mathbf{f},\mathbf{u}} K_{\mathbf{u},\mathbf{u}}^{-1} \mathbf{u}, K_{\mathbf{f},\mathbf{f}} - K_{\mathbf{f},\mathbf{u}} K_{\mathbf{u},\mathbf{u}}^{-1} K_{\mathbf{u},\mathbf{f}}) \overset{\text{DTC}}{\simeq} \mathcal{N}(K_{\mathbf{f},\mathbf{u}} K_{\mathbf{u},\mathbf{u}}^{-1} \mathbf{u}, \mathbf{0}), \tag{4}$$

where $K_{\mathbf{u},\mathbf{u}}$, $K_{\mathbf{f},\mathbf{u}}$ and $K_{\mathbf{u},\mathbf{f}}$ are again shorthand notation for covariance matrices computed by block (e.g. $\text{cov}(f_i, u_j) = \kappa(x_i, z_j)$). The approximation (4) assumes the training conditional covariance $\text{cov}(\mathbf{f}|\mathbf{u})$ is almost $\mathbf{0}$. Marginalization of $u$ from the approximated $p(f_*, \mathbf{f}, \mathbf{u})$ create an

approximation of $p(f_*, \mathbf{f})$. By injecting it into the classic GP framework, we get the DTC posterior

$$p_{\mathrm{DTC}}(f_*|\mathbf{y}) = \mathcal{N}(\sigma_{\mathrm{noise}}^{-2} K_{f_*,\mathbf{u}} S K_{\mathbf{u},\mathbf{f}} \mathbf{y}, \ K_{f_*,f_*} - K_{f_*,\mathbf{u}}(K_{\mathbf{u},\mathbf{u}}^{-1} - S)K_{\mathbf{u},f_*}), \tag{5}$$

where $S = (\sigma_{\mathrm{noise}}^{-2} K_{\mathbf{u},\mathbf{f}} K_{\mathbf{f},\mathbf{u}} + K_{\mathbf{u},\mathbf{u}})^{-1}$. Training has a computational complexity $\mathcal{O}(d^3 N M^2)$ and an additional $\mathcal{O}(d^3 M^2)$ operations are required at each test input $x_*$.

Prediction quality depends on the number of latent variables. Experiments show that using too few of them in large input spaces smooths out the model [12, 13]. Thus, defining $M$ proportionally to the environment size: $M = \rho_z |\Omega|$ is natural. The training and test complexities become respectively $\mathcal{O}(d^3 |\Omega|^3 \rho_x \rho_z^2)$ and $\mathcal{O}(d^3 |\Omega|^2 \rho_z^2)$. The key takeaway is that sparse approximations scale well with increasing input density $\rho_x$, but not so much when the environment size $|\Omega|$ increases. Therefore, it is tempting to model the vector field on a partition of input space [10].

### 2.3  Bayesian Committee Machine

As discussed above, splitting the data and fitting a (sparse) Gaussian process on each subset is natural. It yields a discontinuous model, and prediction quality may be poor around the frontier, even if sub-datasets overlap [11]. To improve upon this naive idea, we can take inspiration from the literature on the aggregation of Gaussian experts Liu et al. [9]. We consider the classic Bayesian Committee Machine (BCM) introduced by Tresp [16].

Let $\mathcal{D}_1, \ldots, \mathcal{D}_J$ be a partition of the dataset $\mathcal{D}$. We assume a Gaussian model, i.e., the prior $p(f_*)$ and the posterior of each expert $p(f_*|\mathcal{D}_i)$ are Gaussians. The BCM introduces a conditional independence approximation between the $\mathcal{D}_i$. Combined with Bayes theorem, it follows that the BCM posterior has the form of a product of expert regularized by a prior term

$$p(\mathcal{D}|f_*) \overset{\mathrm{BCM}}{\simeq} \prod_{i=1}^{J} p(\mathcal{D}_i|f_*) \overset{\mathrm{Bayes}}{\Longrightarrow} p_{\mathrm{BCM}}(f_*|D) \propto \frac{\prod_{i=1}^{J} p(f_*|\mathcal{D}_i)}{p(f_*)^{J-1}} \tag{6}$$

The BCM posterior is Gaussian with closed-form mean and covariance [16]. It is a smooth aggregation that automatically decreases the influence of weak experts (i.e., experts with high uncertainty). Compared to the naive approach, the quality is improved but slower since each prediction requires a prediction from all the experts.

## 3  Method

We present a probabilistic method to learn the magnetic field at a large scale without discontinuities in the prediction from magnetic vectors $y_i \in \mathbb{R}^3$ and their respective positions $x_i \in \mathbb{R}^3$. For that, we split the domain into smaller subdomains. On each, we fit a sparse approximation that we call Gradient-DTC (G-DTC) as described in section 3.1. It embeds prior knowledge from Maxwell equations to reduce the computational complexity and improve the quality of the predictions. G-DTC scales well with increasing spatial densities of the magnetic observation. Then, in section 3.2, we develop a local expert aggregation technique, built upon BCM, that scales with respect to the environment size. Assuming local influence of magnetic data, each local expert in (6) makes a negligible contribution far from its training domain. In our local approximation, we modify the BCM posterior in a principled way so that the contribution of distant experts disappears seamlessly.

### 3.1  Gradient-DTC

Inspired by [10, 18], we wish to inject prior knowledge of magnetic field physics in the DTC formulation to increase the model fidelity and reduce the computational cost. According to Maxwell's equations, the magnetic field is curl-free, provided no free current exists in the domain $\Omega$ [10]. Under such conditions, a scalar potential $\phi$ exists such that $H = -\nabla\phi$. Instead of $\mathbf{u}$ representing magnetic field variables like $\mathbf{f}$, the inter-domain Gaussian process method [19, 17] allows us to mix $f$ and $u$ of different natures. Using this approach, we will manage to do the bulk of the calculation with the scalar latent variables $u_1, \ldots, u_M$, representing $-\phi(z_1), \ldots, -\phi(z_M)$. It will reduce the complexity related to the observation dimension $d$.

By definition, if $f$ and $u$ are two Gaussian processes defined over $\Omega_f$ and $\Omega_u$, and if for every $x_1, \ldots, x_n \in \Omega_f$ and $z_1, \ldots, z_m \in \Omega_u$ the vector $(f(x_1), \ldots, f(x_n), u(z_1), \ldots, u(z_m))$ is Gaussian, then $f$ and $u$ are said to be jointly Gaussian, or to form an inter-domain Gaussian process. It is

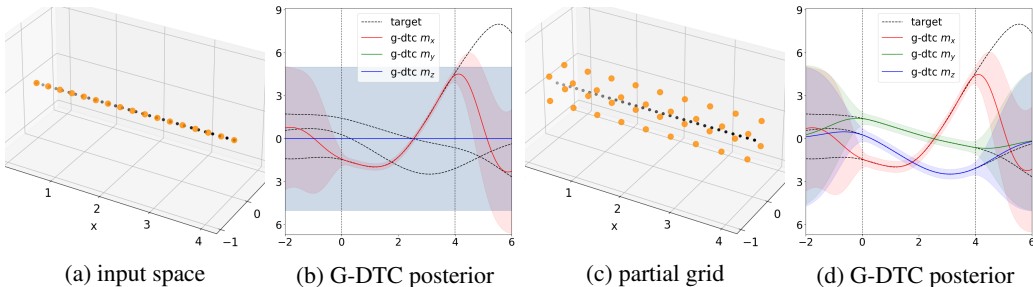

| (a) input space | (b) G-DTC posterior | (c) partial grid | (d) G-DTC posterior |

Figure 1: **Latent inputs degeneracy problem**. (a) Training inputs (black) are placed on the $Ox$ axis. Latent inputs (orange) are spread on this line. (b) With degenerate 1D placement in 3D space, the posterior of the field's $Oy$ and $Oz$ components are equal to their prior ($f_y$ and $f_z$ curves are superimposed). Only the $Ox$ component has been learned. (c-d) With non-degenerated latent inputs, the three components of the posterior are learned properly.

completely defined by their respective mean and kernel function $\mu_f(x), \kappa_f(x, x'), \mu_u(z), \kappa_u(z, z')$, as well as the inter-domain kernel $\kappa_{uf}(z, x) = \kappa_{fu}(x, z)^\top$. [10] suggest to define $u$ as a Gaussian process with differentiable mean $\mu_u = 0$ and twice differentiable kernel:

$$\kappa_u(x, x') = \kappa_{\text{SE}}(x, x') = \sigma_{\text{SE}}^2 \exp\left(-0.5\left(||x - x'||/l_{\text{SE}}\right)^2\right), \tag{7}$$

where $l_{\text{SE}}$ and $\sigma_{\text{SE}}$ are respectively the characteristic length-scale and amplitude of the process. Under such regularity conditions, $f = \nabla u$ is a well defined Gaussian process and $f, u$ are jointly Gaussian [20] with explicit expressions for the mean and covariance kernels

$$\mu_f(x) = \nabla_x \mu_u(x) \qquad\qquad = 0, \tag{8}$$

$$\kappa_{fu}(x, x') = \nabla_x \kappa_u(x, x') \qquad\qquad = -\kappa_{\text{SE}}(x, x')(x - x')/l_{\text{SE}}^2, \tag{9}$$

$$\kappa_f(x, x') = \nabla_x \nabla_{x'}^\top \kappa_u(x, x') \qquad = \kappa_{\text{SE}}(x, x')(I_3 - (x - x')(x - x')^\top/l_{\text{SE}}^2)/l_{\text{SE}}^2. \tag{10}$$

We directly substitute the overloaded kernel $\kappa_u, \kappa_f, \kappa_{fu}$ in (5) to compute the covariance matrices and get what we call G-DTC (G for gradient). Using the overloaded kernel reduces the size of $K_{f_*, \mathbf{u}}$, $K_{\mathbf{f}, \mathbf{u}}$, $K_{\mathbf{u}, \mathbf{f}}$ and $K_{\mathbf{u}, \mathbf{u}}$. Indeed, each block computed from $\kappa_{fu}(x, x')$ is a vector of size $d \times 1$, and $\kappa_u(x, x')$ is a scalar. Compare it against previous block matrices, all computed from $\kappa_f(x, x')$ of size $d \times d$. By mixing vectors of size $d$ with scalars, the new training and test complexities are the same as in a vanilla DTC except that they are proportional to $d = 3$ instead of $d^3 = 27$.

The remaining question is whether the DTC sparse process is a good approximation in this gradient setting. Assuming a differentiable scalar function is perfectly known, then its gradient must also be perfectly known. With enough latent variable $u$, the DTC hypothesis $\text{cov}(\mathbf{f}|\mathbf{u}) \simeq \mathbf{0}$ from (4) should be valid. However, Fig. 1 shows that the posterior is not properly learned when we test this hypothesis on a toy example. In this example, all latent inputs $z_i \in \mathbb{R}^3$ are set along the $Ox$ axis. Therefore, the variations of $u$ are modeled in this direction only. Only the first component of $f = \nabla u$ can be estimated confidently, and it is indeed possible to check that $\text{cov}(\mathbf{f}|\mathbf{u}) \neq 0$. This weakness is shared among state-of-the-art maps using the latent scalar potential [10, 11]. They map the field in 3D even though some of their datasets have positions on a 2D plane.

To solve this issue, we set the latent inputs $z_i$ so that the scalar potential variations are modeled in all directions. The $z_i$ are vertices of a cubic grid of step $\delta = l_{\text{SE}}/2$. We keep only the vertices at a distance $R$ or less from at least one training $x_i$. The parameter $R$ should be as small as possible to reduce the number of latent inputs but large enough such that even isolated $x_i$ have enough $z_i$ in their neighborhood. We set $R = \sqrt{\delta^2 + (\delta/2)^2 + (\delta/2)^2} \simeq 1.23 \cdot \delta$ as justified in the appendix F.

## 3.2 Local Expert Aggregation

As explained previously, the number of latent variables in sparse GP must grow with the domain size. Therefore, we split the domain and fit a G-DTC for each subdomain. Then, we aggregate each local expert prediction with a local version of the BCM. To reduce BCM test runtime, we seek to

consider only experts 'nearby' the prediction location $x_*$ [21, 22], where the 'proximity' is defined by the Gaussian process kernel $\kappa$. The subdomain $\Omega_i$ is 'far' from $x_*$ if $\kappa(x, x_*)$ is small for all $x \in \Omega_i$ [21]. Using the classic squared exponential (SE) kernel (8), we would like to ignore all the experts farther than a distance $l_{\max}$ set to a few $l_{\text{SE}}$.

### 3.2.1 A Local Bayesian Committee Machine

In the BCM framework, it is easy to see from (6) that discarding the prediction of the $i$-th expert is equivalent to replacing the expert posterior by the prior (it cancels out with the denominator). Thus, farther than $l_{max}$ from $\Omega_i$, we approximate the posterior of the $i$-th expert by the prior. Inside $\Omega_i$, we keep the posterior with no further approximations. Everywhere in between, we approximate the posterior of each expert by a geometric mean (GM) of the posterior and the prior

$$p(f_*|\mathcal{D}_i) \simeq p_{\text{GM}}(f_*|\mathcal{D}_i) \propto p(f_*|\mathcal{D}_i)^{\beta(r_i)} p(f_*)^{1-\beta(r_i)}, \tag{11}$$

where $0 \leq \beta(r_i) \leq 1$, and $r_i$ is the distance separating $x_*$ from $\Omega_i$. When $\beta(r_i) = 1$, there is no approximation; when $\beta(r_i) = 0$ the posterior is replaced by the prior. We may drop the argument $r_i$ and note it $\beta_i$ for notional simplicity.

Let $A(x_*) = \{i|\beta(r_i) \neq 0\}$ be the set of index of *active experts*. By injecting the geometric approximation (11) in the BCM posterior (6), we define the Local Bayesian Committee Machine (LBCM) posterior

$$p_{\text{LBCM}}(f_*|\mathcal{D}) \propto \frac{\prod_{i=1}^{J} p(f_*|\mathcal{D}_i)^{\beta_i}}{p(f_*)^{-1+\sum_{i=1}^{J} \beta_i}} = \frac{\prod_{i \in A(x_*)} p(f_*|\mathcal{D}_i)^{\beta_i}}{p(f_*)^{-1+\sum_{i \in A(x_*)} \beta_i}}. \tag{12}$$

Then, as developed in the appendix E, the predictive equations are

$$p_{\text{LBCM}}(f_*|\mathcal{D}) = \mathcal{N}\left(\Lambda^{-1}\left(\sum_{i \in A(x_*)} \beta_i \, \text{cov}(f_*|\mathcal{D}_i)^{-1} \text{E}(f_*|\mathcal{D}_i)\right), \Lambda^{-1}\right) \tag{13}$$

where $\Lambda = (1 - \sum_{i \in A(x_*)} \beta_i)\text{cov}(f_*)^{-1} + \sum_{i \in A(x_*)} \beta_i \, \text{cov}(f_*|\mathcal{D}_i)^{-1}$. The notation $\text{E}(f_*|\mathcal{D}_i)$ and $\text{cov}(f_*|\mathcal{D}_i)$ are for the mean and covariance of the exact posterior $p(f_*|\mathcal{D}_i)$. When all the experts are active, this posterior is precisely what was called the Robust Bayesian Committee Machine (RBCM) in [23]. The difference between our LBCM and their RBCM is the choice of $\beta_i$. Theirs are not functions of the distance $r_i$. They set $\beta_i$ equal to the differential entropy between the prior and the posterior of expert $i$ since their goal was to increase the robustness in the presence of weak experts. On the contrary, we design our $\beta_i$ to save computation time by imposing that its support is known and bounded around $\Omega_i$. We use the cubic hermit spline $\beta(r) = 2(r/l_{\max})^3 - 3(r/l_{\max})^2 + 1$ where $0 \leq r \leq l_{\max}$. It is defined such that it is differentiable, even at the transition points 0 and $l_{\max}$. The optimal $l_{\max}$ depends on your computational budget and quality requirements. In appendix C we illustrate $\beta$ and derive the exact expression of $p_{\text{GM}}$. It is a Gaussian, which allows us to compute its Kullback-Leibler Divergence from the true local expert posterior and justify that $l_{\max} = 2l_{\text{SE}}$ is a reasonable choice.

Our final model defines a G-DTC expert on each of the $J$ subdomains of volume $V$, and it aggregates their predictions with LBCM. The training and test complexities are $\mathcal{O}(d|\Omega|V^2\rho_x\rho_z^2)$ and $\mathcal{O}(d|A(x_*)|V^2\rho_z^2)$ respectively, where $|A(x_*)|$ is the number of active experts. Computing predictions of the magnetic field became independent from the input density and the scene volume.

### 3.2.2 Domain Partition

LBCM is based on BCM, whose performance depends on the domain partition. With smaller subdomains, each expert is faster. However, the conditional independence assumption in (6) is challenged at the frontier between domains. The predictions are better at the interior of a domain, and therefore, the domains must be large enough with respect to $l_{\text{SE}}$. You should also use 'bulky' subdomains to minimize the contact surface. For instance, prefer cubes over flat cuboids. Furthermore, if possible, it is better to set boundaries such that there is little or no data near them. For instance, you could cut horizontally between two floors in a multi-story building. We create a regular partition of input space using boxes of shape $[0, L_1) \times \cdots \times [0, L_{d'})$. Furthermore, we advise that $\min(L_i) > l_{\max}$ so that the number of active experts $|A(x_*)| \leq 2^{d'}$ with equality when $x_*$ is near the corner of a subdomain. In three dimensions $|A(x_*)| <= 8$. Using cuboids makes it straightforward to determine the active experts in constant time and to compute the distances $r$. Details are in the appendix G.

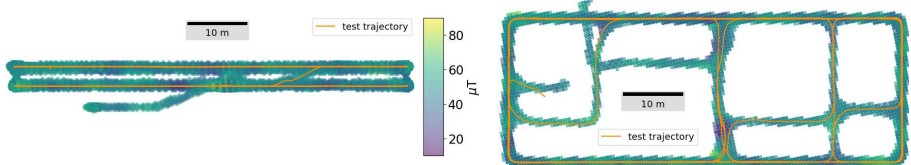

Figure 2: Test trajectory of *Corridor* and the norm of the map learned from the training data

## 4 Experiments

In this section, we compare our approach against the state-of-the-art [10] on both real and simulation data. We first illustrate how G-DTC can take advantage of the spatial distribution of the dataset on a manifold, such as when a robot trajectory is constrained in narrow spaces or limited to 2D navigation on the floor. Then, we show that LBCM combined with G-DTC map efficiently and smoothly the magnetic field of large areas.

### 4.1 Experimental Protocol

**Datasets**. Here we introduce three small simulated datasets called *Simu1D*, *Simu2D* and *Simu3D* illustrated Fig. 4 and a large-scale real one called *Corridor* shown Fig. 2. All datasets are available in our repository https://github.com/CEA-LIST/large-scale-magnetic-mapping.

The simulated datasets are in a $3 \times 3 \times 3\,\mathrm{m}^3$ box, typical of a local expert's domain. Each dataset comprises 1000 noisy training data points and 100 test data points. *Simu1D* is representative of a dataset acquired with motion constrained by the environment, such as in a narrow corridor. All positions are within a tube of $0.5$ meter in diameter. *Simu2D* represents approximately planar motion in open spaces. All positions are within a box of height $0.5$ meter. *Simu3D* represents aerial motion, with positions spread in 3D. Observations are generated jointly from a zero mean Gaussian process as described in more detail in appendix A.

Real data in *Corridor* were collected by the magnetometer and the IMU in an SBG-Ellipse-N rigidly fixed to 4 FLIR Blackfly S cameras. The IMU and the four cameras are used in a graph SLAM [3] to obtain accurate localization of the magnetic observations. The dataset comprises one training and one test trajectory of approximately one kilometer each. They contain 15600 and 16600 observations, respectively, in $1400\,\mathrm{m}^3$ of corridors and staircases in a three-story building. Our model uses a 0 mean prior, so the training observations are preprocessed to subtract their empirical mean, which is then added back to the predictions.

**Metrics**. We evaluate all models with the Mean Squared Error (MSE) and the Mean Standardized Log Loss (MSLL). Both are standard metrics described in [7]. The MSE evaluates how close the posterior mean is from the ground truth. The MSLL also takes into account the quality of the uncertainty. The lower both are, the better the model is. All runtimes are measured on the same laptop from a C++ implementation. We used one core of an 11th Gen Intel® CoreTM i7-11800H with 16 GB of RAM. Prediction times refer to processing the entire test dataset, not just one $x_*$.

**G-DTC And LBCM Setup**. G-DTC depends on the hyper-parameters $l_{\mathrm{SE}}, \sigma_{\mathrm{SE}}$, and $\sigma_{\mathrm{noise}}$. They have the same physical meaning as in a full GP using the covariance kernel (10). Therefore, on *Simu1D*, *Simu2D*, and *Simu3D* we use the available ground truth. Despite our best efforts, unaccounted-for localization errors remain on *Corridor*. We reflected it in the sensor noise by setting $\sigma_{\mathrm{noise}} = 4\,\mu\mathrm{T}$. Then, we randomly selected 1000 training observations in a small area and optimized a GP log marginal likelihood to tune $l_{\mathrm{SE}}$ and $\sigma_{\mathrm{SE}}$. We get $l_{\mathrm{SE}} = 1.35\,\mathrm{m}$ and $\sigma_{\mathrm{SE}} = 6.9\,\mu\mathrm{T}$.

LBCM depends on the domain partition and $l_{\max}$. For reasons that will become clear after the experiment 4.2, we set the side length of the boxes to be equal to $3l_{\mathrm{SE}}$, except when the data covers a multi-story building. In this case, we exploit the natural floor partition and decrease the box height to the height of a story ($\simeq 3$ meter). Finally, we take $l_{\max} = 2l_{\mathrm{SE}}$ as justified section C.

**Competing Method Setup**. Solin et al. [10] introduced a sparse approximation of local experts and a strategy to partially smooth the predictions at the subdomains' boundaries. We abbreviate both contributions G-Hilbert and *Overlap* respectively. *Overlap* requires to define a space partition,

|  | Simu1D | | Simu2D | | Simu3D | |
|---|---|---|---|---|---|---|
|  | *G-DTC* | *G-Hilbert* | *G-DTC* | *G-Hilbert* | *G-DTC* | *G-Hilbert* |
| MSE | **7.7**$\times 10^{-5}$ | $7.8\times 10^{-5}$ | **1.9**$\times 10^{-4}$ | **1.9**$\times 10^{-4}$ | **4.9**$\times 10^{-4}$ | $5.1\times 10^{-4}$ |
| MSLL | -11.8 | **-12.3** | -11.6 | **-11.7** | **-11.1** | -10.9 |
| Fit (ms) | **8.2** | 131 | **29** | 132 | **111** | 135 |
| Predict (ms) | **2.2** | 28 | **8.7** | 28 | 43 | **28** |
| M | 91 | 512 | 211 | 512 | 477 | 512 |

Table 1: Comparison of G-DTC and G-Hilbert using datasets defined near 1D,2D, and 3D manifolds.

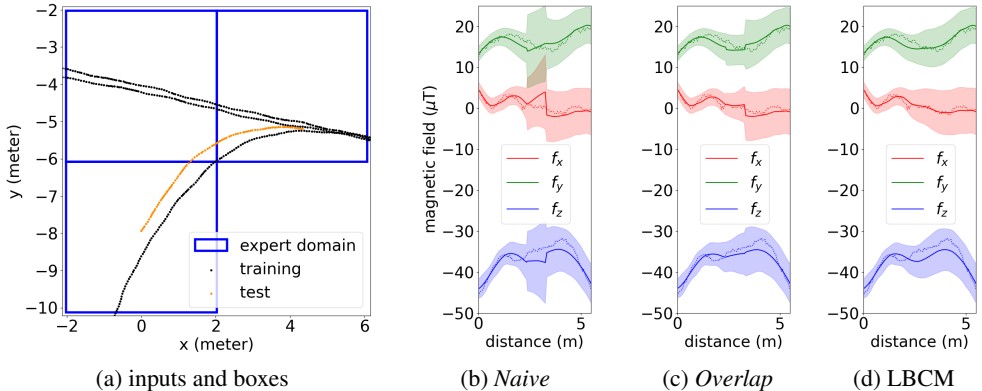

| (a) inputs and boxes | (b) *Naive* | (c) *Overlap* | (d) LBCM |

Figure 3: **Regression discontinuities at the borders**. The dotted lines are the test observations used as ground truth. When the test trajectory crosses domain borders, simple aggregation methods are discontinuous, whereas LBCM is differentiable by definition.

and we use the same as for LBCM. Furthermore, each local expert extends its domain by a range $l_{\mathrm{train}}$ to include training data from neighbor experts, such that the "training boxes" overlap. We set $l_{\mathrm{train}} = 0.5 l_{\mathrm{SE}}$ as in [11]. G-Hilbert relies on basis functions analogous to our latent variables. Three of them are responsible for modeling the unknown mean of the GP prior. In this work, all datasets are preprocessed to be compatible with our 0 mean GP prior assumption. Thus, we remove these three additional basis functions for a fair comparison. Because of mathematical details, G-Hilbert must be defined on a larger domain than the training boxes. We extend each training box by $2l_{\mathrm{SE}}$ as in [11]. Finally, G-DTC and G-Hilbert share the same values for $l_{\mathrm{SE}}, \sigma_{\mathrm{SE}}$, and $\sigma_{\mathrm{noise}}$.

### 4.2 Evaluation of G-DTC Efficiency

In this first experiment, we illustrate G-DTC efficiency when training inputs are located on or near a manifold of lower dimension. For that, we evaluate G-DTC and G-Hilbert on *Simu1D*, *Simu2D*, *Simu3D* and summarize the results in table 1. On *Simu1D*, G-DTC has similar prediction quality and is one order of magnitude faster than G-Hilbert. Increasing the manifold's dimension in *Simu2D* and then in *Simu3D* decreases our speed advantage. In this box of size $3l_{\mathrm{SE}} \times 3l_{\mathrm{SE}} \times 3l_{\mathrm{SE}}$, we could have at most $8 \times 8 \times 8 = 512$ latent inputs. Our worst case is the same as the normal case for G-Hilbert. Suppose we use the same number of latent variables; by looking at the posterior equations, we expect G-DTC to have the same fit time and twice larger prediction time than G-Hilbert. This experiment confirms it on *Simu3D*. For robotic indoor navigation, 1D and 2D input distributions are the most common, which is where our method shines. Notice that we consider the intrinsic dimension and that in this regard, even a trajectory in a staircase is 1D.

### 4.3 Smooth Local Experts Aggregation In Large-Scale Regression

We proved the pertinence of G-DTC on a small domain. Now, we evaluate the scalability and regression quality of LBCM combined with G-DTC. We compare LBCM against *Overlap* and against

|  | G-DTC & *Naive* | G-DTC & *Overlap* | G-DTC & LBCM | G-Hilbert & *Overlap* [10] |
|---|---|---|---|---|
| MSE $(\mu T)^2$ | 1.45 | **1.14** | 1.17 | 1.17 |
| MSLL | **-5.38** | -5.35 | -5.37 | -4.70 |
| Fit (ms) | 65 | 135 | **64** | 4061 |
| Predict (ms) | **108** | 170 | 578 | 4738 |

Table 2: Aggregation methods evaluation in a large building with 16600 test predictions.

the naive aggregation method that uses the space partition as is, without any smoothing strategy. We also compare our full model G-DTC & LBCM against [10] for completeness.

Fig. 2 shows the map built from our model, and table 2 exposes the quantitative results. Surprisingly, the MSE and MSLL are similar for LBCM, *Overlap* and the naive approach. These metrics do not convey the discontinuity issues visible in Fig. 3 that arise at the box boundaries. We see that overlapping training data is insufficient and that LBCM is the only smooth aggregation method. It matters because such discontinuities happen every few meters at each domain transition. Using LBCM comes at the cost of slightly slower predictions because it aggregates a few neighbor experts. On the other hand, *Overlap* is slower to train since overlapping boxes are bigger and contain more inputs and latent inputs. Furthermore, overlapping the domains creates 186 experts instead of 140, with additional experts on the training trajectory's side, top, or bottom. Compared to ours (G-DTC & LBCM), the approach of [10] is two orders of magnitude slower to fit and one order of magnitude slower to predict, mainly because G-Hilbert does not exploit the intrinsic smaller dimension of *Corridor*. Finally, our method combines efficiency with prediction quality and differentiability.

## 5  Discussion

**Limitations**: Our method demonstrated its scalability on large datasets. However, several reasons limit our approach's deployment:

For now, the method fits the map to the data offline. Adding measures incrementally to DTC (called Projected Process in [24, 25]) is possible, and we could extend it to G-DTC.

Furthermore, we assumed that the localization $x$ of each measurement $y$ is perfect, which is false in practice and deteriorates predictions. We solved it by increasing the observation noise, which is an easy fix for model discrepancies [1], but it would be more rigorous to explicitly model input noise [26, 27].

Moreover, we use a low number of latent points to improve runtimes, but it can lead to under-confident G-DTC predictions on dense datasets, especially when the latent point layers are thin, such as in Simu1D and Simu2D.

Another limitation is in the number of new parameters introduced. There is $l_{max}$ and the subdomain size and shape for LBCM, and G-DTC requires a step size $\delta$ and a range $R$ for its partial grid. We described some heuristics to tune them throughout the article, but they remain empirical.

Finally, we describe how to scale the *fit* and *predict* operations on large datasets. Still, we do not provide a way to scale the tuning of our model hyper-parameter $\sigma_{noise}$, $\sigma_{SE}$, and $l_{SE}$. Instead, we learned them from the full GP on a subset of data. It is sensible because they share the same physical meaning for both models, but it is suboptimal.

**Conclusion**: We introduced an efficient approach that creates a smooth map of the magnetic field values and associated uncertainties. We first introduced G-DTC to model the magnetic field, which scales well with respect to increasing data density. It includes physics knowledge from Maxwell equations and can exploit the intrinsic lower dimensions of inputs that naturally arise when collecting measurements along a trajectory. It is faster than other smooth and sparse GP approximations of the magnetic field. We then introduced LBCM to scale with the scene size. Unlike the other approaches, it aggregates local experts in a differentiable way. Efficiency and differentiability are key properties required to develop graph SLAM localization algorithms. For that, online update strategies and robustness to training localization error may be the focus of future research.

**Acknowledgments**

We thank Olivier GOMEZ for its technical support in dealing with the implementation dependencies and cross-platform building. We also thank the reviewers who gave valuable comments to improve the publication's clarity. This work was supported by the CEA List.

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

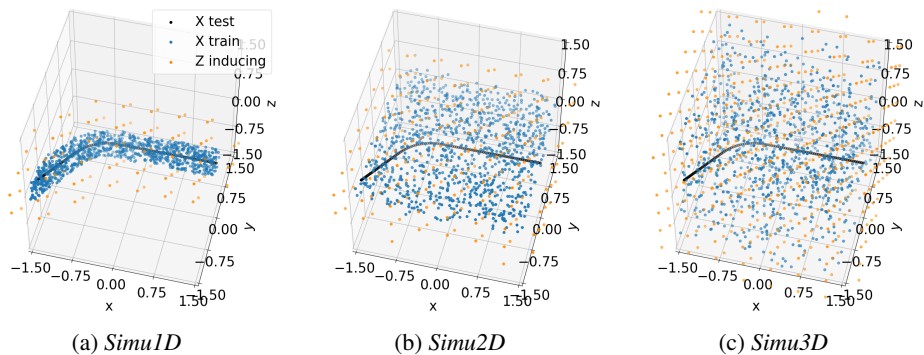

|  |  |  |
|:---:|:---:|:---:|
| (a) *Simu1D* | (b) *Simu2D* | (c) *Simu3D* |

Figure 4: **Simulated dataset and G-DTC** *latent* **inputs**. G-DTC uses a partial grid that creates fewer *latent* inputs (also called *inducing* inputs) on *Simu1D* and *Simu2D* because the training data cover only a fraction of the complete domain.

## A   Simulated Datasets

The three simulated datasets *Simu1D*, *Simu2D*, *Simu3D* displayed in Fig. 4 are, in fact, just one dataset of 3100 data points split into four parts. Each simulated dataset comprises 1000 data points for training, and all share the same 100 test data points. The 3100 observations were sampled jointly from a Gaussian process prior using the kernel (10) where $l_{\text{SE}} = 1$ and $\sigma_{\text{SE}} = 1$. Gaussian white noise was also added to the training observation of each dataset, with $\sigma_{\text{noise}} = 0.1$.

## B   Gaussian Identities

This appendix provides useful identities for manipulating Gaussian distributions and computing the Kullback-Leibler divergence.

Let $p(x) = \mathcal{N}(x|\mu_0, \Sigma_0)$ be the density of a multivariate Gaussian vector of dimension $D$, and $\alpha > 0$ a positive scalar. After re-normalization, the **power** of a Gaussian remains Gaussian

$$\frac{1}{const}p(x)^{\alpha} = \mathcal{N}(x|\mu_0, \frac{1}{\alpha}\Sigma_0). \tag{14}$$

Furthermore, if $q(x) = \mathcal{N}(x|\mu_1, \Sigma_1)$ is another density of a multivariate Gaussian vector, then after re-normalization, the following **product** is Gaussian as well

$$\frac{1}{const}p(x)q(x) = \mathcal{N}(x|\mu_2, \Sigma_2), \tag{15}$$

where

$$\mu_2 = \Sigma_2(\Sigma_0^{-1}\mu_0 + \Sigma_1^{-1}\mu_1), \quad \Sigma_2 = (\Sigma_0^{-1} + \Sigma_1^{-1})^{-1}.$$

Their **quotient** also admits a Gaussian expression

$$\frac{p(x)}{q(x)} \propto \exp\left(-\frac{1}{2}(x - \mu_3)^{\top}\Sigma_3^{-1}(x - \mu_3)\right), \tag{16}$$

where

$$\mu_3 = \Sigma_3(\Sigma_0^{-1}\mu_0 - \Sigma_1^{-1}\mu_1), \quad \Sigma_3 = (\Sigma_0^{-1} - \Sigma_1^{-1})^{-1}.$$

Notice that $\Sigma_3$ can be non-positive, making this quotient unsuitable for a random variable density. However, if $\Sigma_3$ is definite positive, this expression can be normalized into a proper Gaussian density $\mathcal{N}(\mu_3, \Sigma_3)$.

Working with Gaussian densities is convenient since, among many other reasons, their **Kullback-Leibler divergence** admits a closed-form expression

$$D_{KL}(p\,||\,q) = \frac{1}{2}\left(\log\left(\frac{|\Sigma_1|}{|\Sigma_0|}\right) - d + (\mu_0 - \mu_1)^{\top}\Sigma_1^{-1}(\mu_0 - \mu_1) + \text{tr}(\Sigma_1^{-1}\Sigma_0))\right), \tag{17}$$

where the notation $|A|$ and $\text{tr}(A)$ mean respectively the determinant and trace of a matrix $A$.

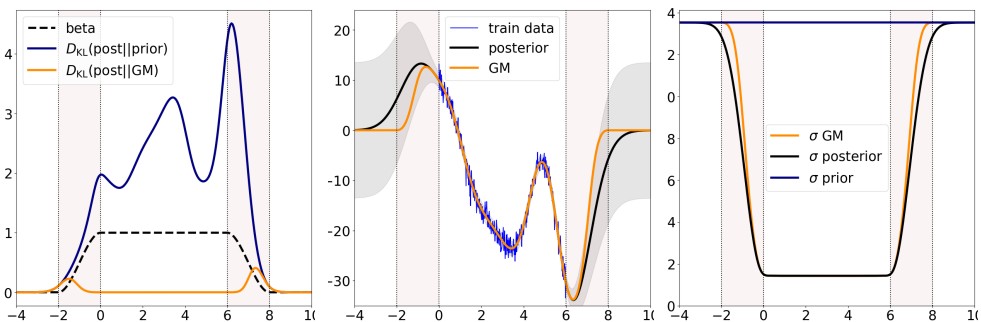

Figure 5: **Illustration of the geometric approximation on a 1D toy dataset**. On all figures, dotted vertical lines delimit the transitions from $\beta = 1$ to $\beta = 0$. The geometric approximation KLD is low everywhere, with a small peak toward the end of the transition zone. Small peaks translate into small posterior mean errors, remaining within the $1\sigma$-error curves of the true posterior. It also translates into higher uncertainties.

## C    Choosing The Beta Function

Section 3.2 we approximated the BCM posterior to introduce LBCM, which happens to have the same posterior as RBCM [23], up to the choice of $\beta$. To the best of our knowledge, it is the first time RBCM is viewed as an approximation of each posterior by a geometric mean between the posterior and prior. We use this theoretical reinterpretation to derive a new metric on the approximation quality with respect to $\beta$.

Assuming the BCM is a good approximation for the full GP posterior, a reasonable and simple way to evaluate the quality of LBCM is to evaluate the approximation separately for each expert. We consider the Kullback-Leibler Divergence (KLD) of the geometric approximation compared to the true posterior: $D_{\mathrm{KL}}(p(f_*|\mathcal{D}_i) \,||\, p_{\mathrm{GM}}(f_*|\mathcal{D}_i))$. To compute it, we need an explicit expression for $p_{\mathrm{GM}}(f_*|\mathcal{D}_i)$. After normalization, identities (14) and (15) tell us that the power of a Gaussian density and the product of two Gaussian densities are also Gaussian. Combining these two results, we get

$$p_{\mathrm{GM}}(f_*|\mathcal{D}_i) = \mathcal{N}\left(\Lambda^{-1}\left(\beta_i \mathrm{cov}(f_*|\mathcal{D}_i)^{-1}\mathrm{E}(f_*|\mathcal{D}_i) + (1-\beta_i)\mathrm{cov}(f_*)^{-1}\mathrm{E}(f_*)\right), \Lambda^{-1}\right) \quad (18)$$

where $\Lambda = \beta_i \mathrm{cov}(f_*|\mathcal{D}_i)^{-1} + (1-\beta_i)\mathrm{cov}(f_*)^{-1}$. Then, the KLD between two Gaussian distributions admits a closed-form expression.

We can now experiment with the $\beta$ function to choose one for which the KLD remains small everywhere. For differentiability, we restrict our-self to the cubic hermit spline $\beta(r) = 2(r/l_{\max})^3 - 3(r/l_{\max})^2 + 1$, where $r$ is the distance separating $x_*$ from the expert domain. $\beta$ is defined up to a hyper-parameter $l_{\max}$ that we will tune experimentally in the remainder of this appendix.

For clarity of the exposition, we start by analyzing a 1D toy example illustrated in Fig 5. The dataset comprises 400 observations on the segment $\Omega_j = [0, 6]$. They are sampled from a Gaussian process prior defined by $\kappa_{\mathrm{SE}}$ and additive white noise. Hyper-parameters are set to $l_{\mathrm{SE}} = 1$, $\sigma_{\mathrm{SE}} = 13.4$ and $\sigma_{\mathrm{noise}} = 1.4$. The KLD between the prior and the posterior is high in the training domain because the posterior incorporates a lot of information from training data. However, at a distance $2l_{\mathrm{SE}}$ from $[0, 6]$, the KLD between the posterior and prior is almost 0, which means the posterior is well approximated by the prior. Therefore, we set $l_{\max} = 2l_{\mathrm{SE}}$, and we verify that the KLD between the posterior and the geometric approximation remains small everywhere.

Then, we conduct the same analysis to tune $l_{\max}$ on real 3D magnetic data. Fig 6 displays the dataset extracted from *Corridor*. We select all the observations made inside a cube of size $4.05 \times 4.05 \times 3$ m$^3$, where $4.05$m $= 3l_{\mathrm{SE}}$ and $3$m is the height of one story in the building. Then, we fit a GP on the training data using the kernel defined in (10), which embeds prior knowledge from magnetic field physics. The second subfigure of Fig. 6 shows that the KLD between the posterior and the prior is almost zero at a distance of $2l_{\mathrm{SE}}$ or more from the training data. Therefore, we use $l_{\max} = 2l_{\mathrm{SE}}$ in the cubic hermit spline formulae given above, which defines the $\beta$ function in the geometric approximation. We then display the KLD between the posterior and the geometric approximation to show that it remains small everywhere, as intended.

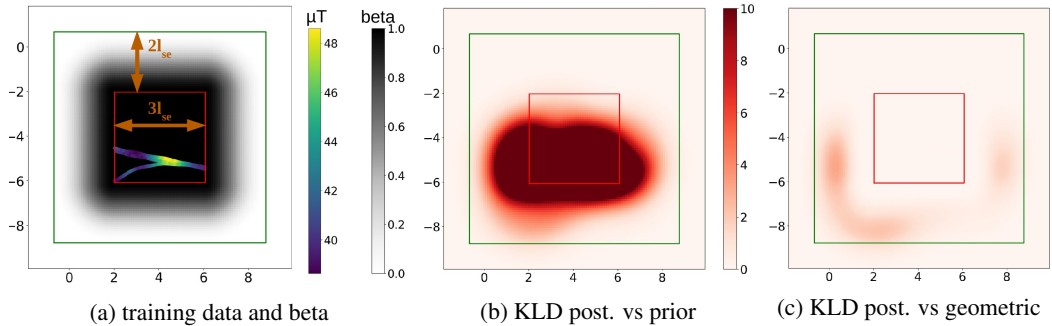

|     |     |     |
| (a) training data and beta | (b) KLD post. vs prior | (c) KLD post. vs geometric |

Figure 6: **Illustration of the geometric approximation on a real dataset**. All figures are top views of the 3D space. The red and green squares delimit the transitions from $\beta = 1$ to $\beta = 0$. The figure on the left displays the training dataset of a local expert, as well as $\beta$. The figure in the middle shows the KLD between the posterior and the prior (saturated to 10 for visualization purposes, but it goes up to 43), which is almost zero at a distance of $2l_{\text{SE}}$ or more from the training data. The figure on the right shows KLD between the posterior and the geometric approximation. It is small everywhere and peaks at 3 toward the end of the transition zone.

## D  Definite Positive Symmetric Matrices

Here, we introduce some well-known results about definite positive symmetric matrices that will be useful to manipulate covariances and derive the LBCM posterior in appendix E. We use the standard notation $S \succ 0$ and $S_1 \succ S_2$ to indicate that the symmetric matrices $S$ and $S_1 - S_2$ are definite positive.

**Theorem 1** *Let $S_1, S_2$ be two symmetric positive matrices with at least one definite positive, and $\alpha_1, \alpha_2$ two strictly positive scalars, then $\alpha_1 S_1 + \alpha_2 S_2 \succ 0$.*

**Theorem 2** *Let $S$ be a symmetric definite positive matrix, then $S^{-1} \succ 0$.*

**Theorem 3** *Let $S_1, S_2$ be two symmetric definite positive matrices, then $S_1 \succ S_2$ if and only if $S_2^{-1} \succ S_1^{-1}$.*

**Theorem 4** *Let $S$ be a symmetric definite positive matrix and $V$ a (possibly rectangular) matrix with full column rank, then $A^\top S A \succ 0$.*

## E  Derivation of LBCM Posterior

In this section, we detail the derivation of the LBMC posterior. It is based on an approximation of the BCM that we can also quickly derive here for completeness.

In general, if we have $J$ experts trained on a partition $\mathcal{D}_1, \ldots, \mathcal{D}_J$ of the dataset $\mathcal{D}$, then from Bayes theorem:

$$p(f_* \mid \mathcal{D}) \overset{\text{Bayes}}{\propto} p(\mathcal{D}|f_*)p(f_*) = p(\mathcal{D}_1, \ldots, \mathcal{D}_J|f_*)p(f_*). \tag{19}$$

The BCM introduces a conditional independence approximation between the $\mathcal{D}_i$, which we inject in (19), and then we use Bayes a second time on each expert posterior

$$p(\mathcal{D}_1, \ldots, \mathcal{D}_J|f_*)p(f_*) \overset{\text{BCM}}{\propto} p(f_*) \prod_{i=1}^{J} p(\mathcal{D}_i|f_*) \overset{\text{Bayes}}{\propto} p(f_*) \prod_{i=1}^{J} \frac{p(f_* \mid \mathcal{D}_i)}{p(f_*)} = \frac{\prod_{i=1}^{J} p(f_* \mid \mathcal{D}_i)}{p(f_*)^{J-1}} \tag{20}$$

Then, we recall that we inject our geometric approximation in (20) to get the LBCM posterior

$$p_{\text{LBCM}}(f_*|\mathcal{D}) \propto \frac{\prod_{i=1}^{J} p(f_*|\mathcal{D}_i)^{\beta_i} p(f_*)^{1-\beta_i}}{p(f_*)^{J-1}} = \frac{\prod_{i \in A(x_*)} p(f_*|\mathcal{D}_i)^{\beta_i}}{p(f_*)^{-1+\sum_{i \in A(x_*)} \beta_i}}. \tag{21}$$

where $A(x_*) = \{i|\beta_i \neq 0\}$. This expression comprises a quotient, products, and powers of Gaussian densities. We straightforwardly combine all the Gaussian identities from appendix B. With a zero mean Gaussian prior, the LBCM posterior has the following exponential form

$$p_{\text{LBCM}}(f_*|\mathcal{D}) \propto \exp\left(-\frac{1}{2}(x-\mu)^\top \Lambda(x-\mu)\right), \tag{22}$$

where

$$\mu = \Lambda^{-1}\left(\sum_{i \in A(x_*)} \beta_i \, \text{cov}(f_*|\mathcal{D}_i)^{-1} \text{E}(f_*|\mathcal{D}_i)\right),$$

$$\Lambda = (1 - \sum_{i \in A(x_*)} \beta_i)\text{cov}(f_*)^{-1} + \sum_{i \in A(x_*)} \beta_i \, \text{cov}(f_*|\mathcal{D}_i)^{-1}. \tag{23}$$

For $p_{\text{LBCM}}$ to be a proper Gaussian density $\mathcal{N}(\mu, \Lambda^{-1})$, all there is left is to check that $\Lambda^{-1}$ is symmetric definite positive. Symmetry is stable by sum, inverse, and multiplication by a scalar, so $\Lambda^{-1}$ is symmetric. Showing positive definiteness is slightly more cumbersome. According to theorem 2, we can directly consider $\Lambda$. We start by rewriting it

$$\Lambda = \text{cov}(f_*)^{-1} + \sum_{i \in A(x_*)} \beta_i \left(\text{cov}(f_*|\mathcal{D}_i)^{-1} - \text{cov}(f_*)^{-1}\right). \tag{24}$$

which is a sum of matrices weighted by positive coefficients that would be positive definite if each matrix is positive definite as well (theorem 1). Using the inversion results (theorem 2), we have $\text{cov}(f_*)^{-1} \succ 0$. And according to 3, the matrix $\text{cov}(f_*|\mathcal{D}_i)^{-1} - \text{cov}(f_*)^{-1}$ is positive definite if and only if $\text{cov}(f_*) \succ \text{cov}(f_*|\mathcal{D}_i)$.

Intuitively, the prior covariance is larger than the posterior one. Thus, $\text{cov}(f_*) \succ \text{cov}(f_*|\mathcal{D}_i)$ should be valid for any sensible expert model, but to finish the formal proof, we need to limit ourself to specific examples. For instance, using full GP experts with kernel $\kappa_{\text{SE}}$,

$$\text{cov}(f_*) - \text{cov}(f_*|\mathcal{D}_i) = K_{\mathbf{f}, f_*}^\top \Sigma K_{\mathbf{f}, f_*}, \tag{25}$$

where $\Sigma = (K_{\mathbf{f},\mathbf{f}} + \sigma_{\text{noise}}^2 I_{nd})^{-1}$. The matrix $K_{\mathbf{f}, f_*}$ has full column rank and using theorem 1 and 2 we have $\Sigma \succ 0$. From theorem 4, it follows that $\text{cov}(f_*) \succ \text{cov}(f_*|\mathcal{D}_i)$ holds for each GP expert and LBCM is well defined. If, instead, we use the DTC or the G-DTC sparse approximation defined from this GP

$$\text{cov}(f_*) - \text{cov}(f_*|\mathcal{D}_i) = K_{\mathbf{u}, f_*}^\top (K_{\mathbf{u},\mathbf{u}}^{-1} - S)K_{\mathbf{u}, f_*}, \tag{26}$$

where $S = (\sigma_{\text{noise}}^{-2} K_{\mathbf{u},\mathbf{f}} K_{\mathbf{f},\mathbf{u}} + K_{\mathbf{u},\mathbf{u}})^{-1}$. We can see that $S^{-1} = \sigma_{\text{noise}}^{-2} K_{\mathbf{u},\mathbf{f}} K_{\mathbf{f},\mathbf{u}} + K_{\mathbf{u},\mathbf{u}} \succ K_{\mathbf{u},\mathbf{u}}$, thus from theorem 3, we have $K_{\mathbf{u},\mathbf{u}}^{-1} - S \succ 0$. It follows from theorem 4 that $\text{cov}(f_*) \succ \text{cov}(f_*|\mathcal{D}_i)$ holds for each experts. Therefore, LBCM is again well-defined.

## F  Partial Grid

G-DTC is based on extracting a partial grid of latent inputs near the training dataset. Here, we give efficient computation techniques for the partial grid, describe some subtleties about latent inputs near the subdomains border, and motivate the chosen value for the parameter $R$.

Given inputs $x_1, \ldots, x_n$ and a radius $R$, we retrieve all the vertices $z_1, \ldots z_M$ from a cubic grid of step $\delta$, that are at a distance $R$ or less from at least one $x_i$. To do so efficiently, for each $x_i$, we generate all the vertices $z$ in a box of center $x_i$ and side length $2R$ (the generation process is described below). Then, we loop through all the vertices of this box to remove those at a distance greater than $R$ from $x_i$. Each remaining $z$ is inserted in $\mathcal{O}(\log(M))$ into a *set* data structure to avoid duplicates. Thus, the partial grid is created in $\mathcal{O}((R/\delta)^3 N \log(M))$ operations, where $(R/\delta)^3$ is proportional to the number of vertices in a ball of radius $R$.

All that is left is to describe the vertices generation process in a box of center $x = (\alpha_1, \ldots, \alpha_{d'})$ and side length $2R$. Let $p_0 = (\gamma_1, \ldots, \gamma_{d'})$ be the origin of the cubic grid of step $\delta$, then it is possible to describe any vertex $z$ by its *grid macro coordinates* $v \in \mathbb{Z}^{d'}$:

$$z = \delta \times v + p_0 \tag{27}$$

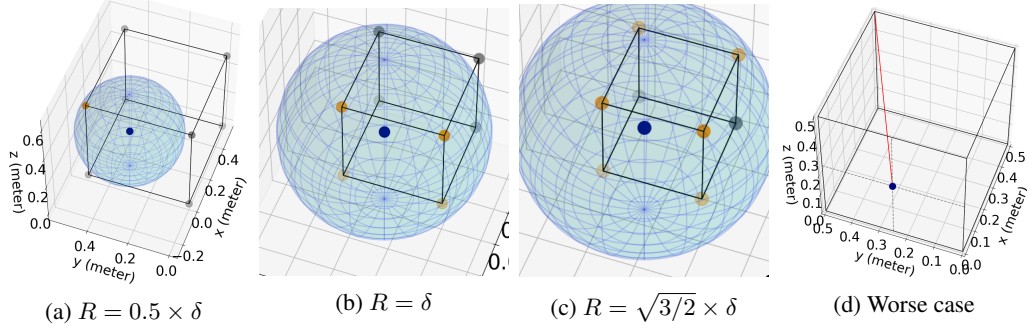

(a) $R = 0.5 \times \delta$    (b) $R = \delta$    (c) $R = \sqrt{3/2} \times \delta$    (d) Worse case

Figure 7: **Vertices selection from one cell of the cubic grid**. The selected vertices (orange) are extracted from a cubic grid (gray) at a distance $R$ or less than an input $x$ (blue). We represent one grid cell only for clarity. (a) $R$ is too small, and only one vertex is selected. (b) $R$ is still too small because the four selected vertices are coplanar (beware, the 4 gray vertices on the back are outside the sphere)(c) $R$ is larger, and the six selected vertices are not coplanar anymore. (d) In the worst case, the input is located in the middle of a face, and $R$ should be greater than the red segment length $\sqrt{\delta^2 + (\delta/2)^2 + (\delta/2)^2} = \sqrt{3/2} \times \delta$.

The lowest and highest macro coordinate values of all the vertices inside the box are respectively

$$
\begin{aligned}
v_{\min} &= \left( \left\lceil \frac{\alpha_1 - R - \gamma_1}{\delta} \right\rceil, \dots, \left\lceil \frac{\alpha_{d'} - R - \gamma_{d'}}{\delta} \right\rceil \right) \\
v_{\max} &= \left( \left\lfloor \frac{\alpha_1 + R - \gamma_1}{\delta} \right\rfloor, \dots, \left\lfloor \frac{\alpha_{d'} + R - \gamma_{d'}}{\delta} \right\rfloor \right)
\end{aligned}
\tag{28}
$$

where $\lceil a \rceil$ and $\lfloor a \rfloor$ denotes the ceiling and integer part of a scalar $a$ respectively. Then $[\![v_{\min,1}, v_{\max,1}]\!] \times \cdots \times [\![v_{\min,d'}, v_{\max,d'}]\!]$ is the list of *grid macro coordinates* of the vertices inside the box.

As a side note, when we split the space $\Omega$ in $\Omega_1, \dots, \Omega_J$ to create the data partition, we allow the partial grid of expert $i$ to go outside $\Omega_i$. In other words, each expert's partial grid can overlap at the subdomain boundaries.

Now we can focus on the ideal value of $R$ in the case of 3D inputs ($d' = 3$). In particular, we want $R$ large enough such that any input $x$ has neighbor vertices spread in 3D, and we say that $R = \sqrt{\delta^2 + (\delta/2)^2 + (\delta/2)^2}$ is the smallest of such values that work for any input $x$. Fig. 7 illustrates it.

## G  Box Partition

For any given input $x$, LBCM is based on efficiently retrieving to which expert domain $x$ belongs and which are the neighbor experts. We must also compute the distance $r$ from $x$ to any neighbor expert domains. A regular partition of the space into boxes is convenient to compute these quantities in constant time.

Let $c = (\gamma_1, \dots, \gamma_{d'})$ be the center of a box of shape $[0, L_1) \times \cdots \times [0, L_{d'})$. Then its distance from any $x = (\alpha_1, \dots, \alpha_{d'})^\top$ is

$$
r = \sqrt{\sum_{i=1}^{d'} \max\left( |\alpha_i - \gamma_i| - \frac{L_i}{2}, 0 \right)^2}
\tag{29}
$$

where the notation $|a|$ stands for the absolute value of a scalar $a$.

Each box can be identified by its *macro coordinates* $b \in \mathbb{Z}^{d'}$, which represents how many boxes should be skipped in each direction from an *origin box* at $b = (0, \dots, 0)$, defined arbitrarily. If $c$ is the center of the *origin box*, then the macro coordinates of the box containing $x$ are

$$
b(x) = \left( \left\lfloor \frac{\alpha_1 - \gamma_1}{L_1} \right\rfloor, \dots, \left\lfloor \frac{\alpha_{d'} - \gamma_{d'}}{L_{d'}} \right\rfloor \right).
\tag{30}
$$

where $\lfloor a \rfloor$ denotes the integral part of $a$.

To get all the neighbor boxes that intersect the ball of center $x$ and radius $l$, we start by finding all boxes that intersect the box of center $x$ and shape $[0, l) \times \cdots \times [0, l)$. It creates a small list of candidate boxes. Then we loop through all of them, compute the distance $r$ according to (29), and keep only the ones such that $r \leq l$.

All that is left is to compute the candidates list. For that, we compute $b(x - \lambda)$ and $b(x + \lambda)$ where $\lambda = (l, \ldots, l)$. Then $[\![b(x - \lambda)_1, b(x + \lambda)_1]\!] \times \cdots \times [\![b(x - \lambda)_{d'}, b(x + \lambda)_{d'}]\!]$ is the list of candidate *macro coordinates*, where the double brackets $[\![a, b]\!]$ stands for the set of integers between $a$ and $b$ included.

