# OpenReview forum: "Large Scale Mapping of Indoor Magnetic Field by Local and Sparse Gaussian Processes"
_robot-learning.org/CoRL/2024/Conference — CoRL 2024_

### Official Review · Reviewer_MA22 · 2024-07-03
**A valuable contribution to robotics research for large scale mapping, applicable beyond magnetic field mapping.**

**Originality:** 3
**Technical Quality:** 4
**Clarity Of Presentation:** 5
**Potential Impact:** 4
**Recommendation:** 3
**Confidence:** 4

**Review:**

This is a well written paper with clear structure and presentation of ideas and results.

The paper is significant as it contributes to multiple expert modelling and physics-based learning techniques. The concepts in this paper of Gradient-DTC (G-DTC) and Local Bayesian Committee Machines (LBCM) can be applied outside of the application presented (magnetic field mapping) and to other mapping problems. e.g. G-DTC may prove useful for mapping any curl-free field via sparse Gaussian processes, LBCM may be used in any probabilistic/Bayesian multiple-expert learning framework. Discontinuities at borders of submaps is a common hurdle in large-scale mapping problems in robotics, and LBCM is naturally differentiable in these regions, lending itself well to the problem.

Strengths:
- strong theoretical contributions that are applicable to a broad range of mapping problems
- well formulated paper
- evaluation on both simulated data and on a real dataset collected by the authors
- demonstrated computational advantage compared to state-of-the-art methods
- competitive mapping accuracy and uncertainty estimation with state-of-the-art methods

Weaknesses:
- no explicit limitations section and only minor discussion of limitations in text
- a stronger robotics application would be an extension to online mapping as a physical robot collects measurements with a magnetometer.

**Quality Of The Limitations Section:**

1

**Questions For Rebuttal:**

- The corridor dataset the authors have collected could be another valuable contribution to research -- have you considered making it and the details of its collection available?
- The choice of the $l_{max}$ for the beta function as discussed in Appendix B does not seem rigorous. Fig 5. does demonstrate that $2l_{SE}$ is suitable, but is this true of all datasets and partitions? How transferrable is this analysis to higher dimensional problems and other domains?
- Beyond $l_{max}$, hyperparameter choice and sensitivity in general could be discussed more. i.e. how the domain is partitioned, are boxes always recommended? Does this change with domain shape?

Minor comments:
- Broken reference to appendix in line 198 of manuscript
- "two one order of magnitude slower" in line 279
- should refer to table 1 in line 256
- mathcal Ox missing in first line of Figure 1 caption

**Robotics Focus:**

3

**Summary Of Paper:**

This paper presents a methodology for large scale mapping of indoor magnetic fields. The application to robotics is the use-case of magnetometer-based navigation and localisation, which require a magnetic map of the environment a priori. The methodology is based on a physics-informed sparse Gaussian process regression framework over multiple localised domains, with a novel expert aggregation technique to recover large scale modelling from the local regression. Aggregation is performed such that the result is differentiable and smooth. The entire framework demonstrates computational advantage compared to state-of-the-art alternatives, with competitive mapping accuracy and uncertainty estimation.

**Summary Of Recommendation:**

I believe this is a good theoretical contribution to addressing mapping problems via mixed-expert learning frameworks, and to incorporating physics-based knowledge into sparse Gaussian process regression. Uncertainty in my recommendation comes from a lack of explicit discussion regarding the limitations of the methodology. Some may also believe this paper has a weaker focus on learning for a physical robotics problem -- it is evaluated on real data but there is no true "hardware" involved in this problem, though it is certainly a problem relevant to robotics..

---

### Official Review · Reviewer_KGGy · 2024-07-19
**Interesting research problem, good solution that improves on state of the art, unclear limitations**

**Originality:** 4
**Technical Quality:** 4
**Clarity Of Presentation:** 4
**Potential Impact:** 3
**Recommendation:** 3
**Confidence:** 3

**Review:**

The paper addresses an important problem with an original approach, where the infamous intractability of Gaussian Processes is intelligently addressed. The exposition is sufficiently clear given the page limit and the required mathematical background. I recommend making the source code available online, as the method might contain non-obvious implementation details, for reproducibility. The experimental campaign is convincing, although it would have been nice to see an experiment comparing the current approach vs a traditional mapping approach to corroborate the claims made in the introduction. Without this experiment, I am not sure this paper will convince people to try this method instead of just tweaking their favorite traditional mapping library.

I could not find the limitations section, which should have been included (see the official call for papers https://www.corl.org/contributions/call-for-papers). For example, one major limitation is that the current method might not work well online (which is briefly mentioned in the conclusions).


Minor comments:

1. Missing reference to the appendix in Section 3.2.2
2. Fix the citation at Line 95
3. Wrong citation in the last column of Table 2?

**Quality Of The Limitations Section:**

1

**Questions For Rebuttal:**

1. What are the limitations of the work?

**Robotics Focus:**

3

**Summary Of Paper:**

This paper presents a method for large scale mapping of indoor magnetic fields. The method includes splitting the domain into subdomains, fitting sparse approximations on them, and a local expert aggregation technique. The approach is validated on real and simulated data.

**Summary Of Recommendation:**

I did not go through the Appendix, but the paper seems solid, hence my recommendation.

---

### Official Review · Reviewer_XHZx · 2024-07-21
**Exposition seems to need a major revision**

**Originality:** 2
**Technical Quality:** 2
**Clarity Of Presentation:** 1
**Potential Impact:** 2
**Recommendation:** 2
**Confidence:** 4

**Review:**

Strenghts:
- The topic of magnetic mapping is interesting and has potential to complement visual SLAM.
- There are no severe grammar or spelling mistakes in the paper.
- Ideas of leveraging local GPs along with inducing point approximation seems sensible and worthwhile to try.

Weakness:

- (major) The paper does not distinguish existing ideas properly. The paper needs related work section to locate the contributions within the state-of-the-art.

For example, embedding maxwell equation into GP prior is an existing idea [1]. Leveraging only few local neighboring experts for more efficient local GP aggregation is also a known idea [2]. There are several ways to handle discontinuity of local GPs, e.g., [3-4] and the paper misses to differentiate itself from existing works clearly. I think a thourough related work section, especially with a focus on new techniques, can help the paper.

- (major) The paper writing can improve significantly.

It was difficult to parse section 3.1 and 3.2. I could only understanding by reading [5], where several figures and intuition are prsented clearly. Looking back, section 3.1 alone will be difficult to comprehend by many readers. Secondly, the connection from the mapping to the use of GPs is poorly written. There should be one subsection devoted instead of a sentence in section 3. Lastly, a direct derivation of equation 12 and 13 should be presented in the paper or supplementary.

- (minor) Table 1 and table 2 seem to need standard deviations rather than only reporting the mean values.

[1] Modeling magnetic fields using gaussian processes
[2] Trust Your Robots! Predictive Uncertainty Estimation of Neural Networks with Sparse Gaussian Processes
[3] Efficient Computation of Gaussian Process Regression for Large Spatial Data Sets by Patching Local Gaussian Processes
[4] Patchwork kriging for large-scale gaussian process regression
[5] Modeling and interpolation of the ambient magnetic field by gaussian processes

**Quality Of The Limitations Section:**

1

**Questions For Rebuttal:**

One of the key advantage of local GPs is also in the data uncertainty term. Instead of having one constant noise, each local GP can represent more complex noise terms, which intuitively, will make more sense in mapping applications (each magnetometer might depending on the environment produce different noise characteristics). Why not exploit such characteristics?

**Robotics Focus:**

3

**Summary Of Paper:**

This paper focuses on magnetic maps of indoor environments, which can be used in indoor positioning and navigation applications. Here, magnetic field is measured by magnetometers for example. Main contributions of this paper are twofolds. First, the paper modifies a Determinsitic Training Conditional spasre GP to include prior knowledge from Maxwell equations. Second, the paper modifies aggregation methods of local Gaussian Processes (e.g., an ensemble of GPs that divides input space) for improved efficiency and continuous predictions. To create large scale map of indoor magnetic field, the paper combines these two techniques. Simulation and real dataset are used to evaluate the proposed methodologies.

**Summary Of Recommendation:**

My recommendation is a weak reject due to the quality of the presentation. I can guess what is meant, and I could look up related literature to guess what the contributions are, but having to guess make it difficult to judge the main contribution of the paper.

---

### Author Rebuttal · Authors · 2024-08-09

**Limitation section**: We would like to apologize for this missing section. We missed this guideline because it was our first time submitting an article at CoRL. We will include the following limitations in the final version:
1. For now, the method **fit the map to the data offline**. However, we are working toward an online version. Adding measures incrementally to DTC (called Projected Process in [1]) has already been done, and we could extend it to G-DTC.
2. We made the **hypothesis that the localization x of each measurement y is perfect**, which is not verified in practice. It yields over-confident predictions. We solved it by increasing the observation noise. It is a classic and easy way to deal with model discrepancies [2], but a more rigorous treatment would be to explicitly model input noise [3,4].
3. We use a low number of inducing points to improve runtimes. However, experiments showed that it leads to **under-confident G-DTC predictions** on dense datasets, especially when the inducing point layers are thin, such as in Simu1D and Simu2D.
4. Another limitation is in the **number of new parameters introduced**. There is $l_\text{max}$ and the subdomain size and shape for LBCM. Moreover, G-DTC requires a step size $\delta$ and a range $R$ for its partial grid. We described some heuristics to tune them throughout the article, but they remain empirical.
5. Finally, our article describes how to scale the fit and predict operations on large datasets. Still, **we do not provide a way to scale the Gaussian Process hyper-parameter tuning** ($\sigma_\text{noise}$, $\sigma_\text{se}$, and $l_\text{se}$). Instead, as mentioned in section 4.3, we learned them from the full GP on a subset of data. It makes sense because they have the same physical meaning for both models, but it is suboptimal.

**The other answers are in the “official comment” of each reviewer.**


[1] Csató, Lehel, and Manfred Opper. "Sparse on-line Gaussian processes." Neural computation, 2002

[2] M. Kok and A. Solin. “Scalable Magnetic Field SLAM in 3D Using Gaussian Process Maps”. In 21st International Conference on Information Fusion (FUSION), 2018

[3] McHutchon, Andrew, and Carl Rasmussen. "Gaussian process training with input noise." Advances in neural information processing systems, 2011.

[4] Damianou, Andreas C., Michalis K. Titsias, and Neil D. Lawrence. “Variational inference for latent variables and uncertain inputs in Gaussian processes”. 2016.

---

### Decision · Program_Chairs · 2024-09-04

**Decision:**

Accept

**Comment:**

Summary of Strengths:
* Magnetic mapping is a relevant topic.
* The concepts presented are well formulated (G-DTC and LBCM) can be applied to other problems.
* The contribution lies in the multiple expert modelling and physics-based learning techniques.
* Evaluation on both simulated data and a real dataset collected by the authors.
* Demonstrated computational advantage and mapping accuracy (including uncertainty) compared to state-of-the-art methods.

Summary of Weaknesses:
* Limitations are not discussed adequately -> this has been addressed in the rebuttal and needs to be added to the paper
* The paper needs a proper related work section, where it differentiates clearly from other state-of-the-art approaches. Noting that similar concepts have been presented already in the literature.  -> A Related Work section should be added (some space can be saved in Section 2).
* Readability. -> to be addressed in the paper
* Reproducibility of the results. -> to be addressed as per discussion in the rebuttal